# Expanding the stdpopsim species catalog, and lessons learned for realistic genome simulations

M Elise Lauterbur[1]*, Maria Izabel A Cavassim[2†], Ariella L Gladstein[3†], Graham Gower[4†], Nathaniel S Pope[5†], Georgia Tsambos[6†], Jeffrey Adrion[5,7], Saurabh Belsare[5], Arjun Biddanda[8], Victoria Caudill[5], Jean Cury[9], Ignacio Echevarria[10], Benjamin C Haller[11], Ahmed R Hasan[12,13], Xin Huang[14,15], Leonardo Nicola Martin Iasi[16], Ekaterina Noskova[17], Jana Obsteter[18], Vitor Antonio Correa Pavinato[19], Alice Pearson[20,21], David Peede[22,23], Manolo F Perez[24], Murillo F Rodrigues[5], Chris CR Smith[5], Jeffrey P Spence[25], Anastasia Teterina[5], Silas Tittes[5], Per Unneberg[26], Juan Manuel Vazquez[27], Ryan K Waples[28], Anthony Wilder Wohns[29], Yan Wong[30], Franz Baumdicker[31], Reed A Cartwright[32], Gregor Gorjanc[33], Ryan N Gutenkunst[34], Jerome Kelleher[30], Andrew D Kern[5], Aaron P Ragsdale[35], Peter L Ralph[5,36], Daniel R Schrider[37], Ilan Gronau[38]*

[1]Department of Ecology and Evolutionary Biology, University of Arizona, Tucson, United States; [2]Department of Ecology and Evolutionary Biology, University of California, Los Angeles, Los Angeles, United States; [3]Embark Veterinary, Inc, Boston, United States; [4]Section for Molecular Ecology and Evolution, Globe Institute, University of Copenhagen, Copenhagen, Denmark; [5]Institute of Ecology and Evolution, University of Oregon, Eugene, United States; [6]School of Mathematics and Statistics, University of Melbourne, Melbourne, Australia; [7]Ancestry DNA, San Francisco, United States; [8]54Gene, Inc, Washington, United States; [9]Universite Paris-Saclay, CNRS, INRIA, Laboratoire Interdisciplinaire des Sciences du Numerique, Orsay, France; [10]School of Life Sciences, University of Glasgow, Glasgow, United Kingdom; [11]Department of Computational Biology, Cornell University, Ithaca, United States; [12]Department of Cell and Systems Biology, University of Toronto, Toronto, Canada; [13]Department of Biology, University of Toronto Mississauga, Mississauga, Canada; [14]Department of Evolutionary Anthropology, University of Vienna, Vienna, Austria; [15]Human Evolution and Archaeological Sciences (HEAS), University of Vienna, Vienna, Austria; [16]Department of Evolutionary Genetics, Max Planck Institute for Evolutionary Anthropology, Leipzig, Germany; [17]Computer Technologies Laboratory, ITMO University, St Petersburg, Russian Federation; [18]Agricultural Institute of Slovenia, Department of Animal Science, Ljubljana, Slovenia; [19]Entomology Department, The Ohio State University, Wooster, United States; [20]Department of Genetics, University of Cambridge, Cambridge, United Kingdom; [21]Department of Zoology, University of Cambridge, Cambridge, United Kingdom; [22]Department of Ecology, Evolution, and Organismal Biology, Brown University, Providence, United States; [23]Center for Computational Molecular Biology, Brown University, Providence, United States; [24]Department of Genetics and Evolution, Federal University of Sao Carlos, Sao Carlos, Brazil; [25]Department of Genetics, Stanford University School of Medicine, Stanford, United States; [26]Department of Cell and Molecular Biology, National Bioinformatics Infrastructure Sweden, Science for Life Laboratory, Uppsala

*For correspondence: lauterbur@email.arizona.edu (MEL); ilan.gronau@runi.ac.il (IG)

†These authors contributed equally to this work

University, Uppsala, Sweden; [27]Department of Integrative Biology, University of California, Berkeley, Berkeley, United States; [28]Department of Biostatistics, University of Washington, Seattle, United States; [29]Broad Institute of MIT and Harvard, Cambridge, United States; [30]Big Data Institute, Li Ka Shing Centre for Health Information and Discovery, University of Oxford, Oxford, United Kingdom; [31]Cluster of Excellence - Controlling Microbes to Fight Infections, Eberhard Karls Universit¨at Tubingen, Tubingen, Germany; [32]School of Life Sciences and The Biodesign Institute, Arizona State University, Tempe, United States; [33]The Roslin Institute and Royal (Dick) School of Veterinary Studies, University of Edinburgh, Edinburgh, United Kingdom; [34]Department of Molecular and Cellular Biology, University of Arizona, Tucson, United States; [35]Department of Integrative Biology, University of Wisconsin–Madison, Madison, United States; [36]Department of Mathematics, University of Oregon, Eugene, United States; [37]Department of Genetics, University of North Carolina at Chapel Hill, Chapel Hill, United States; [38]Efi Arazi School of Computer Science, Reichman University, Herzliya, Israel

**Abstract** Simulation is a key tool in population genetics for both methods development and empirical research, but producing simulations that recapitulate the main features of genomic datasets remains a major obstacle. Today, more realistic simulations are possible thanks to large increases in the quantity and quality of available genetic data, and the sophistication of inference and simulation software. However, implementing these simulations still requires substantial time and specialized knowledge. These challenges are especially pronounced for simulating genomes for species that are not well-studied, since it is not always clear what information is required to produce simulations with a level of realism sufficient to confidently answer a given question. The community-developed framework stdpopsim seeks to lower this barrier by facilitating the simulation of complex population genetic models using up-to-date information. The initial version of stdpopsim focused on establishing this framework using six well-characterized model species (Adrion et al., 2020). Here, we report on major improvements made in the new release of stdpopsim (version 0.2), which includes a significant expansion of the species catalog and substantial additions to simulation capabilities. Features added to improve the realism of the simulated genomes include non-crossover recombination and provision of species-specific genomic annotations. Through community-driven efforts, we expanded the number of species in the catalog more than threefold and broadened coverage across the tree of life. During the process of expanding the catalog, we have identified common sticking points and developed the best practices for setting up genome-scale simulations. We describe the input data required for generating a realistic simulation, suggest good practices for obtaining the relevant information from the literature, and discuss common pitfalls and major considerations. These improvements to stdpopsim aim to further promote the use of realistic whole-genome population genetic simulations, especially in non-model organisms, making them available, transparent, and accessible to everyone.

## eLife assessment

This **important** paper reports recent improvements and extensions to stdpopsim, a community-driven resource that is built on top of powerful software for performing simulations of population genomic data and provides a catalog of species with curated genomic parameters and demographic models. In addition to describing the new features and species in stdpopsim, the authors provide a set of practical guidelines for implementing realistic simulations. Overall, this **convincing** manuscript serves as an excellent overview of the utility, challenges, common pitfalls, and best practices of population genomic simulations. It will be of broad interest to population, evolutionary, and ecological geneticists studying humans, model organisms, or non-model organisms.

## Introduction

Population genetics allows us to answer questions across scales from deep evolutionary time to ongoing ecological dynamics, and dramatic reductions in sequencing costs enable the generation of unprecedented amounts of genomic data that can be used to address these questions (*Ellegren, 2014*). Ongoing efforts to systematically sequence life on Earth by initiatives such as the Earth Biogenome (*Lewin et al., 2022*) and its affiliated project networks, such as Vertebrate Genomes (*Rhie et al., 2021*), 10,000 Plants (*Cheng et al., 2018*) and others (*Darwin Tree of Life Project Consortium, 2022*), are providing the backbone for enormous increases in the amount of population-level genomic data available for model and non-model species. These data are being used, among other things, in the inference of population history and demographic parameters (*Beichman et al., 2018*), studying adaptive introgression (*Gower et al., 2021*), providing null expectations for selection scans (e.g. *Hsieh et al., 2021*), and understanding the implications of deleterious variation in populations of conservation concern (e.g. *Robinson et al., 2023*). While many of the methods that address these questions were initially developed for a few key model systems such as humans and *Drosophila*, more recent efforts are generalizing these methods to include important factors not initially accounted for, such as inbreeding or selfing (*Blischak et al., 2020*), skewed offspring distributions (*Montano, 2016*), and intense artificial selection even for non-model organisms (*MacLeod et al., 2013*; *MacLeod et al., 2014*).

Simulations can be useful at all stages of this work—for planning studies, analyzing data, testing inference methods, and validating findings from empirical and theoretical research. For instance, simulations provide training data for inference methods based on machine learning (*Schrider and Kern, 2018*) and Approximate Bayesian Computation (*Csilléry et al., 2010*). They can also serve as baselines for further analyses: for example, simulations incorporating demographic history serve as null models when detecting selection (*Hsieh et al., 2016*) or seed downstream breeding program simulations (*Gaynor et al., 2021*). More recently, population genomic simulations have been used to help guide conservation decisions for threatened species (*Teixeira and Huber, 2021*; *Kyriazis et al., 2022*).

Increasing amounts of data and the sophistication of inference methods have enabled researchers to ask ever more specific and precise questions. Consequently, simulations must incorporate more and more elements of biological realism. Important elements include genomic features such as mutation and recombination rates that strongly affect genetic variation and haplotype structure (*Nachman, 2002*). The inclusion of these genomic features is particularly important when linked selection is acting upon the patterns of genomic diversity being studied (*Cutter and Payseur, 2013*). Furthermore, the demographic history of a species—encompassing population sizes and distributions, divergences, and gene flow—can dramatically affect patterns of genomic variation (*Teshima et al., 2006*). Thus species-specific estimates of these and other ecological and evolutionary parameters (such as those governing the process of natural selection) are important when generating realistic simulations. This presents challenges, especially to new researchers, as it takes a great deal of specialized knowledge not only to code the simulations themselves but also to find and choose appropriate estimates of the parameters underlying the simulation model.

The recently developed community resource stdpopsim provides easy access to detailed population genomic simulations (*Adrion et al., 2020*). It lowers the technical barriers to performing these simulations and reduces the possibility of erroneous implementation of simulations for species with published demographic models. The initial release of stdpopsim was restricted to only six well-characterized model species, such as *Drosophila melanogaster* and *Homo sapiens*, but the feedback we received from the community identified a widespread desire to simulate a broader range of non-model species, and ideally to incorporate these into the stdpopsim catalog for future use. This feedback, and subsequent efforts to expand the catalog, also uncovered a vital need to better understand when it is practical to create a realistic simulation of a species of interest, and indeed what 'realistic' means in this context.

This paper reports on the updates made in the current release of stdpopsim (version 0.2), and is also intended as a resource for any researcher who wishes to develop chromosome-scale simulations for their own species of interest. We start by describing the central idea behind the standardized simulation framework of stdpopsim, and then outline the main updates made to the stdpopsim catalog and simulation framework in the past two years. We then provide guidelines for generating

population genomic simulations, either for the purpose of using them in one specific study, or with the intent of making the simulations available for future work by adding the appropriate models to stdpopsim. Among other considerations, we discuss when a chromosome-scale simulation is more useful than simulations based on either individual loci or generic loci. We specify the required input data, mention common pitfalls in choosing appropriate parameters, and suggest courses of action for species that are missing estimates of some necessary inputs. We conclude with examples from two species recently added to stdpopsim, which demonstrate some of the main considerations involved in the process of designing realistic chromosome-scale simulations. While the guidelines provided in this paper are intended for any researcher interested in implementing a population genomic simulation using any software, we highlight the ways in which the stdpopsim framework eases the burden involved in this process and facilitates reproducible research.

## The utility of stdpopsim for chromosome-scale simulations

We begin by providing a brief overview of the importance of chromosome-scale simulations and the main rationale behind stdpopsim; see *Adrion et al., 2020* for more on the topic. The main objective of population genomic simulations is to recreate patterns of sequence variation along the genome under the inferred evolutionary history of a given species. To achieve this, stdpopsim is built on top of the msprime (*Kelleher et al., 2016*; *Nelson et al., 2020*; *Baumdicker et al., 2021*) and SLiM (*Haller and Messer, 2019*) simulation engines, which are capable of producing fairly realistic patterns of sequence variation if provided with accurate descriptions of the genome architecture and evolutionary history of the simulated species. The required parameters include the number of chromosomes and their lengths, mutation and recombination rates, the demographic history of the simulated population, and, potentially, the landscape of natural selection along the genome. A key challenge when setting up a population genomic simulation is to obtain estimates of all of these quantities from the literature and then correctly implement them in an appropriate simulation engine. Detailed estimates of all of these quantities are increasingly available due to the growing availability of population genomic data coupled with methodological advances. Incorporating this data into a population genomic simulation often involves integrating this data between different literature sources, which can require specialized knowledge of population genetics theory. Thus, the process of coding a realistic simulation can be quite time-consuming and often error-prone.

The main objective of stdpopsim is to streamline this process, and to make it more robust and more reproducible. Contributors collect parameter values for their species of interest from the literature and then specify these parameters in a template file for the new model. This model then undergoes a peer-review process, which involves another researcher independently recreating the model based on the provided documentation. Automated scripts then execute to compare the two models; if discrepancies are found in this process, they are resolved by discussion between the contributor and reviewer, and if necessary with the input of additional members of the community. This quality-control process quite often finds subtle bugs (e.g. as in *Ragsdale et al., 2020*) or highlights parts of the model that are ambiguously defined by the literature sources. This increases the reliability and reproducibility of the resulting simulations in any downstream analysis.

Another important goal of stdpopsim is to promote and facilitate chromosome-scale simulations, as opposed to the common practice of simulating many short segments (see, e.g. *Harris and Nielsen, 2016*). Simulation of long sequences, on the order of $10^7$ bases, has until recently been computationally prohibitive, but this has changed with the development of modern simulation engines such as msprime and SLiM. Generating chromosome-scale simulations has several key benefits. First, the organization of genes on chromosomes is a key feature of a species' genome that is ignored in many traditional population genomic simulations (see *Schrider, 2020* for one exception). Second, modeling physical linkage allows simulations to capture important correlations between genetic variants on a chromosome. These correlations reduce variance relative to separate and independent simulations of equivalent genetic material. This has a particularly striking effect in long stretches with a low recombination rate, as observed for instance on the long arm of human chromosome 22 (*Dawson et al., 2002*). In bacteria, a similar effect occurs due to genome-wide linkage that is broken only by the horizontal transfer of short segments (*Didelot and Maiden, 2010*). When conducting simulations with natural selection, genetic linkage has an even stronger effect. Selection acting on a small number of sites can indirectly influence levels and patterns of genetic variation at linked neutral sites, which has

been shown to have a widespread effect on patterns of genomic variation in a myriad of species (e.g. *McVicker et al., 2009*; *Charlesworth, 2012*). In addition, the lengths of chromosome-scale shared haplotypes within and between populations provide valuable information on their demographic history. Demographic inference methods that use such information, such as MSMC (*Schiffels and Wang, 2020*) and IBDNe (*Browning and Browning, 2015*), perform best on long genomic segments with realistic recombination rates. Chromosome-scale simulations are clearly required to test (or train) such methods, or to conduct power analyses when designing empirical studies that use them. With stdpopsim, such simulations are available with just a single call to a command-line script or with the execution of a handful of lines of Python code.

## Additions to stdpopsim

When first published, the stdpopsim catalog included six species: *Homo sapiens*, *Pongo abelii*, *Canis familiaris*, *Drosophila melanogaster*, *Arabidopsis thaliana*, and *Escherichia coli* (*Figure 1*). One way the catalog has expanded is through the introduction of additional demographic models for *Homo sapiens*, *Pongo abelii*, *Drosophila melanogaster*, and *Arabidopsis thaliana*, enabling a wider variety of simulations for these well-studied species. However, the initial collection of six species represents only a small slice of the tree of life. This is a concern not only because there is a large community of researchers studying other organisms, but also because methods developed for application to model species (such as humans) may not perform well when applied to other species with very different biology. Adding species to the stdpopsim catalog will allow developers to easily test their methods across a wider variety of organisms.

We thus made a concerted effort to recruit members of the population and evolutionary genetics community to add their species of interest to the stdpopsim catalog. This effort involved a series of workshops to introduce potential contributors to stdpopsim, followed by a 'Growing the Zoo' hack-athon organized alongside the 2021 ProbGen conference. The seven initial workshops allowed us to reach a broad community of more than 150 researchers, many of whom expressed interest in adding non-model species to stdpopsim. The hackathon was then structured based on feedback from these participants. One month before the hackathon, we organized a final workshop to prepare interested participants, by introducing them to the process of developing a new species model and adding it to the stdpopsim code base. Roughly 20 scientists participated in the hackathon (most of whom are included as authors on this paper), which resulted in the addition of 15 species to the stdpopsim catalog (*Figure 1*). The catalog now includes a teleost fish (*Gasterosteus aculeatus*), a bird (*Anas platyrhynchos*), a reptile (*Anolis carolinensis*), a livestock species (*Bos taurus*), six insects including two vectors of human disease (*Aedes aegypti* and *Anopheles gambiae*), a nematode (*Caenorhabditis elegans*), two flowering plants including a crop (*Helianthus annuus*), an algae (*Chlamydomonas rein-hardtii*), two bacteria, four primates, and a common mammalian associate of humans (*Canis familiaris*). Not all of these have recombination maps or demographic models (see *Figure 1*), but this lays a framework for future contributions.

Expanding the species catalog required adding several capabilities to the simulation framework of stdpopsim. Some features were added by upgrading the neutral simulation engine, msprime, from version 0.7.4 to version 1.0 (*Baumdicker et al., 2021*). Among other features, this upgrade includes a discrete-site model of mutation, which enables simulating sites with multiple mutations and possibly more than two alleles. Another key feature added to stdpopsim's simulation framework was the ability to model non-crossover recombination. In bacteria and archaea, genetic material can be exchanged through horizontal gene transfer, which can add new genetic material (e.g. via the transfer of plas-mids) or replace homologous sequences through homologous recombination (*Thomas and Nielsen, 2005*; *Didelot and Maiden, 2010*; *Gophna and Altman-Price, 2022*). However, the initial version of stdpopsim used crossover recombination to stand in for these processes. Although we cannot currently simulate varying gene content (as would be required to simulate the addition of new genetic material by horizontal gene transfer), the msprime and SLiM simulation engines now allow gene conversion, which has the same effect as non-crossover homologous recombination. Following *Cury et al., 2022*, we use this to include non-crossover homologous recombination in bacterial and archaeal species. This is done in stdpopsim by setting a flag in the species model to indicate that recombination should be modeled without crossovers and specifying an average tract length of exchanged genetic material. For example, the model for *Escherichia coli* has been updated in the stdpopsim catalog to use

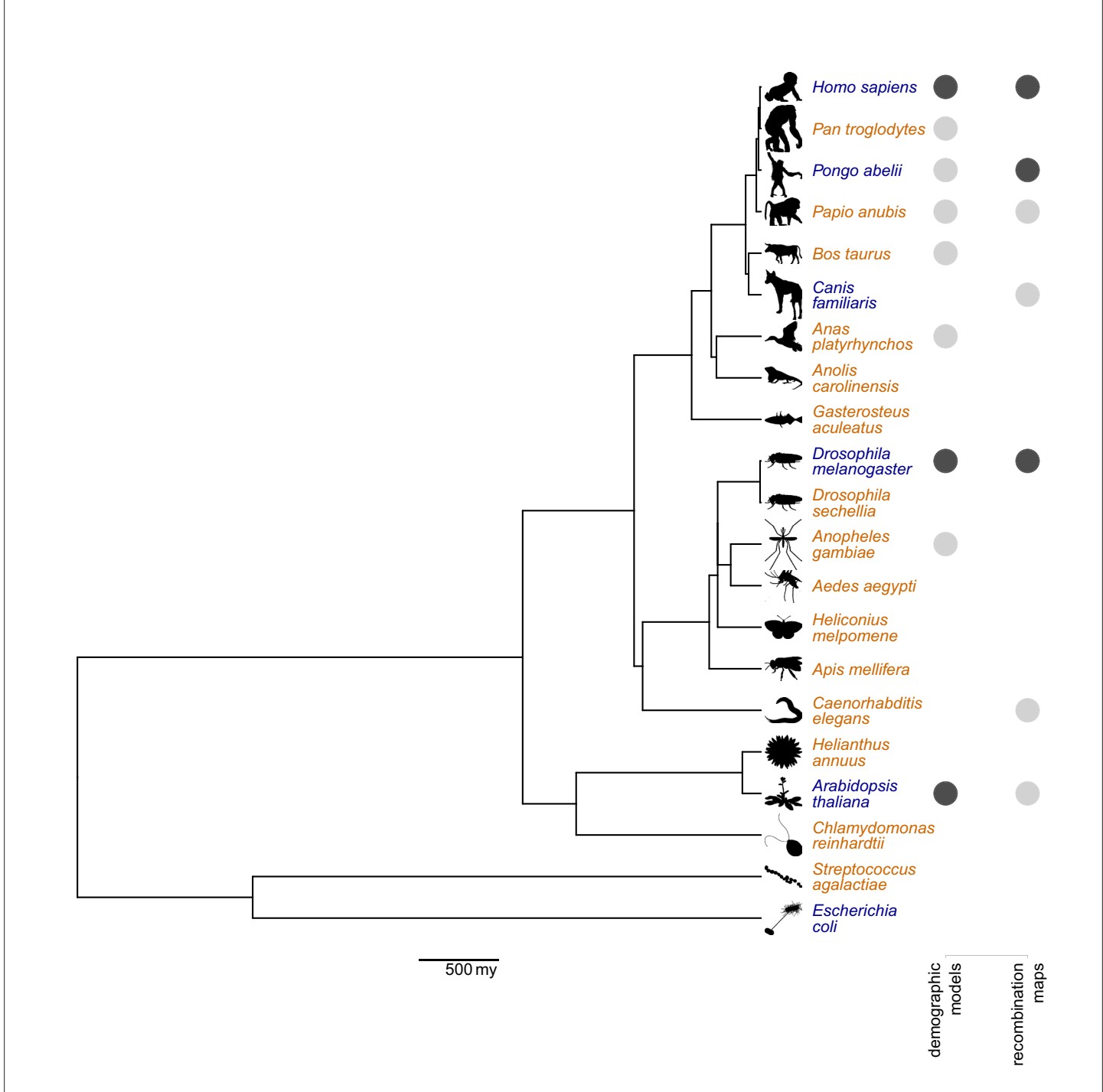

**Figure 1.** Phylogenetic tree of species available in the stdpopsim catalog, including the six species we published in the original release (**Adrion et al., 2020**, in blue), and 15 species that have since been added (in orange). Solid circles indicate species that have one (light gray) or more (dark gray) demographic models and recombination maps. Branch lengths were derived from the divergence times provided by TimeTree5 (**Kumar et al., 2022**). The horizontal bar below the tree indicates 500 million years (my). Source code for generating the tree is given in **Figure 1—source code 1 and 2**.

The online version of this article includes the following source code for figure 1:

**Source code 1.** R code for generating the figure.

**Source code 2.** Newick-format tree used as input in R code.

non-crossover recombination at an average rate of $8.9 \times 10^{-11}$ recombination events per base per generation, with an average tract length of 542 bases (*Wielgoss et al., 2011*; *Didelot et al., 2012*). Note that this rate ($8.9 \times 10^{-11}$) corresponds to the rate of initiation of a recombined tract.

Recombination without crossover is also prevalent in sexually reproducing species, where it is termed *gene conversion*. Gene conversion affects shorter segments than crossover recombination and creates distinct patterns of genetic diversity along the genome (*Korunes and Noor, 2017*). Indeed, gene conversion rates in some species are estimated to occur at similar or even higher rates than crossover recombination (*Gay et al., 2007*; *Comeron et al., 2012*; *Wijnker et al., 2013*). To accommodate this in stdpopsim simulations, one needs to specify the fraction of recombinations that occur due to gene conversion (i.e. without crossover), and the average tract length. For example, the model for *Drosophila melanogaster* has been updated in the stdpopsim catalog to have a fraction of gene conversions of 0.83 (in all chromosomes that undergo recombination) and an average tract length of 518 bases (*Comeron et al., 2012*). This update does not affect the rate of crossover recombination, but it adds gene conversion events at a ratio of 83:17 relative to crossover recombination events. We note that since non-crossover recombination incurs a high computation load in simulation, it is turned off by default in stdpopsim, and must be explicitly invoked by the simulation model. Note that ignoring gene conversion may result in a slightly skewed distribution of shared haplotypes between individuals (see Table 1).

Another important extension of stdpopsim allows augmenting a genome assembly with genome annotations, such as coding regions, promoters, and conserved elements. These annotations can be used to simulate selection at a subset of sites (such as the annotated coding regions) using parametric distributions of fitness effects. Standardized, easily accessible simulations that include the reality of pervasive linked selection in a species-specific manner has long been identified as a goal for evolutionary genetics (e.g. *McVicker et al., 2009*; *Comeron, 2014*). Thus, we expect this extension of stdpopsim to be transformative in the way simulations are carried out in population genetics. This significant new capability of the stdpopsim library will be detailed in a forthcoming publication, and is not the focus of this paper.

## Guidelines for implementing a population genomic simulation

The concentrated effort to add species to the stdpopsim catalog has led to a series of important insights about this process, which we summarize here as a set of guidelines for implementing realistic simulations for any species. Our intention is to provide general guidance that applies to any population genomic simulation software, but we also mention specific requirements that apply to simulations done in stdpopsim.

## Basic setup for chromosome-level simulations

Implementing a realistic population genomic simulation for a species of interest requires a detailed description of the organism's demography and mechanisms of genetic inheritance. While simulation

---

**Table 1.** Guidelines for dealing with missing parameters.
For each parameter, we provide a suggested course of action, and mention the main discrepancies between simulated data and real genomic data that could be caused by misspecification of that parameter.

| Missing parameter | Suggested action | Possible discrepancies |
|---|---|---|
| Mutation rate | Borrow from the closest relative with a citable mutation rate | Number of polymorphic sites |
| Recombination rate | Borrow from the closest relative with a citable recombination rate | Patterns of linkage disequilibrium |
| Gene conversion rate and tract length | Set the rate to 0 or borrow from the closest relative with a citable rate | Lengths of shared haplotypes across individuals |
| Demographic model | Set the effective population size ($N_e$) to a value that reflects the observed genetic diversity | Features of genetic diversity that are captured by the site frequency spectrum, such as the prevalence of low-frequency alleles |

software requires unforgivingly precise values, in practice we may only have rough guesses for most of the parameters describing these processes. In this section, we list the relevant parameters and provide guidelines for how to set them based on current knowledge.

1. A chromosome-level genome assembly, which consists of a list of chromosomes or scaffolds and their lengths. Having a good quality assembly with complete chromosomes, or at least very long scaffolds, is necessary if chromosome-level population genomic simulations are to reflect the genomic architecture of the species. When expanding the stdpopsim catalog during the 'Growing the Zoo' hackathon, we considered the possibility of adding species whose genome assemblies are composed of many relatively small contigs, unanchored to chromosome-level scaffolds. Although we had not previously put restrictions on which species might be added, we decided that we would only add species with chromosome-level assemblies. The main justification for this restriction is that species with less complete genome builds typically do not have good recombination maps and demographic models, making chromosome-level simulation much less useful in such species. Another issue is the storage burden and long load times involved in dealing with hundreds of contigs. Finally, each species requires validation of its code before it is added to the stdpopsim catalog, as well as long-term maintenance to keep it up-to-date with changes made to the stdpopsim framework. So, the benefit of including species with very partial genome builds in stdpopsim would be outweighed by the substantial extra burden on stdpopsim maintainers as well as downstream users of these models. Another reason to focus on species with chromosome-level assemblies is that we expect their numbers to dramatically increase in the near future due to numerous genome initiatives (*Lewin et al., 2022*; *Rhie et al., 2021*; *Cheng et al., 2018*) and the development of new long-read sequencing technologies and assembly pipelines (*Chakraborty et al., 2016*; *Amarasinghe et al., 2020*; *Amarasinghe et al., 2021*).

2. An average mutation rate for each chromosome (per generation per bp). This rate estimate can be based on sequence data from pedigrees, mutation accumulation studies, or comparative genomic analysis calibrated by fossil data (i.e. phylogenetic estimates). At present, stdpopsim simulates mutations at a constant rate under the Jukes–Cantor model of nucleotide mutations (*Jukes and Cantor, 1969*). However, we anticipate future development will provide support for more complex, heterogeneous mutational processes, as these are easily specified in both the SLiM and msprime simulation engines. Such progress will further improve the realism of simulated genomes, since mutation processes, including rates, are known to vary along the genome and through time (*Benzer, 1961*; *Ellegren et al., 2003*; *Supek and Lehner, 2019*).

3. Recombination rates (per generation per bp). Ideally, a population genomic simulation should make use of a chromosome-level recombination map, since the recombination rate is known to vary widely across chromosomes (*Nachman, 2002*), and this can strongly affect the patterns of linkage disequilibrium and shared haplotype lengths. When this information is not available, we suggest specifying an average recombination rate for each chromosome. At a minimum, an average genome-wide recombination rate needs to be specified, which is typically available for well-assembled genomes. For bacteria and archaea, which primarily experience non-crossover recombination, the average tract length should also be specified (see details in the previous section). Gene conversion (optional): If one wishes to model gene conversion in eukaryotes, either together with crossover recombination or as a stand-alone process, then one should specify the fraction of recombinations done by gene conversion as well as the per chromosome average tract length.

4. A demographic model describing ancestral population sizes, split times, and migration rates. Selection of a reasonable demographic model is often crucial, since misspecification of the model can generate unrealistic patterns of genetic variation that will affect downstream analyses (e.g. *Navascués and Emerson, 2009*). A given species might have more than one demographic model, fit from different data or by different methods. Thus, when selecting a demographic model, one should examine the data sources and methods used to obtain it to ensure that they are relevant to the study at hand (see also Limitations of simulated genomes below). At a minimum, simulation requires a single estimate of effective population size. This estimate, which may correspond to some sort of historical average effective population size, should produce simulated data that matches the average observed genetic diversity in that species. Note, however, that this average effective population size cannot capture features of genetic variation that are caused by recent changes in population size and the presence of population structure (*MacLeod et al., 2013*; *Eldon et al., 2015*). For example, a recent population expansion will produce an excess of low-frequency alleles that no simulation of a constant-sized population will reproduce (*Tennessen et al., 2012*).

5. An average generation time for the species. This parameter is an important part of the species' natural history. This value does not directly affect the simulation, since stdpopsim uses either the Wright–Fisher model (in SLiM) or the Moran model (in msprime), both of which operate in time units of generations. Thus, the average generation time is only currently used to convert time units to years, which is useful when comparing among different demographic models.

These five categories of parameters are sufficient for generating simulations under neutral evolution. Such simulations are useful for a number of purposes, but they cannot be used to model the influence of natural selection on patterns of genetic variation. To achieve this, the simulator needs to know which regions along the genome are subject to selection, and the nature and strength of this selection. As mentioned above, the ability to simulate chromosomes with realistic models of selection is still under development, and will be finalized in the next release of stdpopsim. The development version of stdpopsim enables simulation with selection (using the SLiM engine) by specifying genome annotations and distributions of fitness effects, as specified below.

6. Genome annotations, specifying regions subject to selection (as, for example, a GFF3/GTF file). For instance, annotations can contain information on the location of coding regions, the position of specific genes, or conserved non-coding regions. Regions not covered by the annotation file are assumed to be evolving free from the effects of direct natural selection.

7. Distributions of fitness effects (DFEs) for each annotation. Each annotation is associated with a DFE describing the probability distribution of selection coefficients (deleterious, neutral, and beneficial) for mutations occurring in the region covered by the annotation. DFEs can be inferred from population genomic data (reviewed in *Eyre-Walker and Keightley, 2007*), and are available for several species (e.g. *Ma et al., 2013*; *Huber et al., 2018*).

The current release of stdpopsim contains annotations and implemented DFE models for the three model species: *A. thaliana*, *D. melanogaster*, and *H. sapiens*. A forthcoming publication will provide details about how this is implemented in stdpopsim and examples of possible uses of this feature.

## Extracting parameters from the literature

Simulations cannot of course precisely match reality, but in setting up simulations it is desirable to choose parameters that best reflect our current understanding of the evolutionary history of the species of interest. In practice, a researcher may choose each parameter to match a fairly precise estimate or a wild guess, which may be obtained from a peer-reviewed publication or by word of mouth. However, values in stdpopsim are always chosen to match published estimates, so that the underlying data and methods are documented and can be validated. Because the process of converting information reported in the literature to parameters used by a simulation engine is quite error-prone, independent validation of the simulation code is crucial. We highly recommend following a quality-control procedure similar to the one used in stdpopsim, in which each species or model added to the catalog is independently recreated or thoroughly reviewed by a separate researcher.

Obtaining reliable and citable estimates for all model parameters is not a trivial task. Oftentimes, values for different parameters must be gleaned from multiple publications and combined. For example, it is not uncommon to find an estimate of a mutation rate in one paper, a recombination map in a separate paper, and a suitable demographic model in a third paper. Integrating information from different publications requires caution, since some of these parameter estimates are entangled in non-trivial ways. For instance, consider simulating a demographic model estimated in a specific paper that assumes a certain mutation rate. Naively using the demographic model, as published, with a new estimate of the mutation rate will lead to levels of genetic diversity that do not fit the genomic data. This is addressed in stdpopsim by allowing a demographic model to be simulated using a mutation rate that differs from the default rate specified for the species. See, for example, the model implemented for *Bos taurus*, which is described in the next section. This important feature does not necessarily fix all potential inconsistencies caused by assumptions made by the demographic inference method (such as assumptions about recombination rates). It is therefore recommended, when possible, to take the demographic model, mutation rates, and recombination rates from the same study, and to proceed carefully when mixing sources. An additional tricky source of inconsistency is coordinate drift between subsequent versions of genome assemblies. In stdpopsim, we follow the UCSC Genome Browser and use liftOver to convert the coordinates of recombination maps and genome annotations to the coordinates of the current genome assembly (*Hinrichs et al., 2006*).

## Limitations of simulated genomes

Despite their great utility, simulated genomes cannot fully capture all aspects of genetic variation as observed in real data, with some aspects modeled better than others. As mentioned above, this will strongly depend on the demographic model used in the simulation. Thus, it is important to consider the potential limitations of different demographic models in reflecting observed genetic variation. First, a demographic model inferred from analysis of genomic data will likely depend on the samples that contributed the analyzed genomes. The inferred demographic model can only reflect the genealogical ancestry of these sampled individuals, and this will typically make up a small portion of the complete genealogical ancestry of the species. Thus, demographic models inferred from larger sets of samples from diverse ancestry backgrounds may potentially provide a more comprehensive depiction of genetic variation within a species. This is true if sufficiently realistic demographic models can be fit—models that account for the structure of populations within a species. That said, the choice of samples used for inference will mostly influence recent changes in genetic variation. This is because the genealogy of even a single individual consists of numerous ancestors in each generation in the deep past, which is the premise of methods that infer ancestral population sizes from a single input genome (*Li and Durbin, 2011*).

The computational method used for inference also affects the way genetic variation is reflected by the demographic model, because different methods derive their inference from different features of genomic variation. Some methods make use of the site frequency spectrum at unlinked single sites (e.g. *Gutenkunst et al., 2009*; *Excoffier et al., 2013*; *Liu and Fu, 2015*), while other methods use haplotype structure (e.g. *Li and Durbin, 2011*; *Schiffels and Wang, 2020*; *Browning and Browning, 2015*). This, in turn, may influence the accuracy of different features in the inferred demography. For example, very recent demographic changes, such as recent admixture or bottlenecks, are difficult to infer from the site frequency spectrum, but are more easily inferred by examining shared long haplotypes (as demonstrated by the demographic model inferred for *Bos taurus* by *MacLeod et al., 2013*; see below). Several studies have compared different approaches to demographic inference (e.g. *Harris and Nielsen, 2013*; *Beichman et al., 2017*), but unfortunately, there is currently no succinct handbook that describes the relative strengths and weaknesses of different methods. Thus, assessing the potential limitations of a given demographic model currently requires some familiarity with the method used for its inference. In addition, all methods assume that the input sequences are neutrally evolving. This implies that technical choices, such as the specific genomic segments analyzed and various filters, may also influence the inferred model and its ability to model observed genetic variation. Thus, it is strongly advised to read the study that inferred the demographic model and understand potential limitations that stem from the selection of samples, methods, and filters.

We note that the inclusion of a demographic model in the stdpopsim catalog does not involve any judgment as to which aspects of genetic variation it captures. Any model that is a faithful implementation of a published model inferred from genomic data can be added to the stdpopsim catalog. Thus, potential users of stdpopsim should use the implemented models with the appropriate caution, keeping in mind the limitations discussed above. We maintain a fairly detailed documentation page for the catalog (see Data availability), which contains a brief summary for each demographic model. This summary includes a graphical description of the model (such as the one shown for *Anopheles gambiae* in Figure 2B), as well as a description of the data and method used for inference. Therefore, the documentation can provide guidance to potential users of stdpopsim in the process of selecting an appropriate demographic model for simulation. Finally, we hope that the standardized simulations implemented in stdpopsim will facilitate additional studies that examine the relative strengths and limitations of different approaches to demographic inference (and modeling genetic variation in general), and this will allow us to generate even more realistic simulations in the future.

## Filling in the missing pieces

For many species, it is difficult to obtain estimates of all necessary model parameters. *Table 1* provides suggestions for ways to deal with missing values of various model parameters. The table also mentions possible consequences of the misspecification of each parameter.

In some cases, one may wish to generate simulations for a species with a partial genome build. Despite the focus of stdpopsim on species with chromosome-level assemblies (see discussion above), simulation is still potentially useful for species with less complete assemblies, with some important

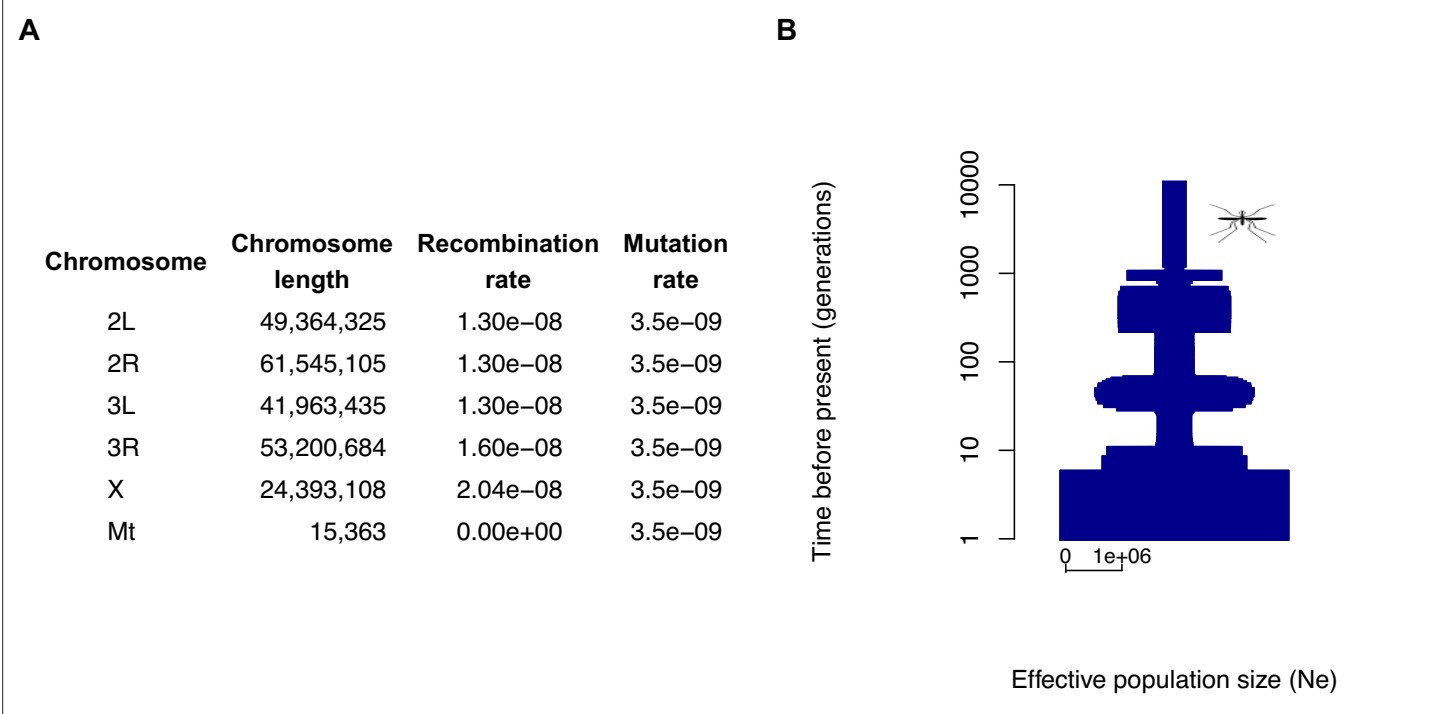

**Figure 2.** The species parameters and demographic model used for *Anopheles gambiae* in the stdpopsim catalog. (**A**) The parameters associated with the genome build and species, including chromosome lengths, average recombination rates (per base per generation), and average mutation rates (per base per generation). (**B**) A graphical depiction of the demographic model, which consists of a single population whose size changes throughout the past 11,260 generations in 67-time intervals (note the log scale). The width at each point depicts the effective population size ($N_e$), with the horizontal bar at the bottom indicating the scale for $N_e = 10^6$. This figure is adapted from the data on the stdpopsim catalog documentation page (see Data availability) and plotted with POPdemog (*Zhou et al., 2018*). Source code for generating the figure is given in *Figure 2—source code 1*.

The online version of this article includes the following source code for figure 2:

**Source code 1.** R code for generating the figure.

considerations to keep in mind. Longer contigs or scaffolds in these builds can be simulated separately and independently. This approach allows us to model genetic linkage within each contig, but genetic linkage between different contigs that map to the same chromosome will not be captured by the simulation. This provides a reasonable approximation for many purposes, at least for genomic regions far from the contig edges. For shorter contigs, separate independent simulations will not be able to capture patterns of long-range linkage in a reasonably realistic way. Thus, a potentially viable option for shorter contigs is to combine them into longer pseudo-chromosomes, trying to mimic the species' expected chromosome lengths. Despite their somewhat artificial construction, these pseudo-chromosomes have the important benefit of capturing patterns of linkage similar to those observed in real genomic chromosomes. If, for example, the main purpose of the simulation is to examine the distribution of lengths of shared haplotypes between individuals, or study patterns of background selection, then it makes sense to simulate such pseudo-chromosomes. However, genetic correlations between different specific contigs lumped together in this way are obviously not accurate. So, if the main purpose of the simulation is to examine local patterns of genetic variation in loci of interest, then it may be more appropriate to simulate the relevant contigs separately (even if they are short), or to randomly sample several mappings of contigs to pseudo-chromosomes. For some purposes, it makes sense to simulate a large number of unlinked sites (*Gutenkunst et al., 2009*; *Excoffier et al., 2013*), which can be generated without any sort of genome assembly. However, this approach would not have the benefits of chromosome-scale simulations. While some of the same considerations hold when simulating unlinked short sequences, a detailed discussion about such simulations goes beyond the scope of this paper. Ultimately, the recommended mode of simulation for a species with a partial genome assembly depends on the intended use of the simulated genomes.

## Examples of added species

In this section, we provide examples of two species recently added to the stdpopsim catalog, *Anopheles gambiae,* and *Bos taurus*, to demonstrate some of the key considerations of the process. In each example, we highlight in bold the model parameters set for each species.

### Anopheles gambiae (mosquito)

*Anopheles gambiae*, the African malaria mosquito, is a non-model organism whose population history has direct implications for human health. Several large-scale studies in recent years have provided information about the population history of this species on which population genomic simulations can be based (e.g. *Miles, 2017*; *Clarkson et al., 2020*). The genome assembly structure used in the species model is from the AgamP4 **genome assembly** (*Sharakhova et al., 2007*), downloaded directly from Ensembl (*Howe et al., 2020*) using the special utilities provided by stdpopsim.

Estimates of average **recombination rates** for each of the chromosomes (excluding the mitochondrial genome) were taken from a recombination map inferred by *Pombi et al., 2006* which itself included information from *Zheng et al., 1996*; *Figure 2A*. As direct estimates of **mutation rate** (e.g. via mutation accumulation) do not currently exist for *Anopheles gambiae*, we used the genome-wide average mutation rate of $\mu = 3.5 \times 10^{-9}$ mutations per generation per site estimated by *Keightley et al., 2009* for the fellow Dipteran *Drosophila melanogaster*, a rate that was used for the analysis of *A. gambiae* data in *Miles, 2017*. To obtain an estimate for the default **effective population size** ($N_e$), we used the formula $\theta = 4\mu N_e$, with the above mutation rate ($\mu = 3.5 \times 10^{-9}$ mutations per base per generation) and a mean nucleotide diversity of $\theta \approx 0.015$, as reported by *Miles, 2017* for the Gabon population. This resulted in an estimate of $N_e = 1.07 \times 10^6$, which we rounded down to one million. These steps were documented in the code for the stdpopsim species model, to facilitate validation and future updates. We acknowledge that some of these steps involve somewhat arbitrary choices, such as the choice of the Gabon population and rounding down of the final value. However, this should not be seen as a considerable source of misspecification, since this value of $N_e$ is meant to provide only a rough approximation to historical population sizes and would be overwritten by a more detailed demographic model. *Miles, 2017* inferred demographic models from *Anopheles* samples from nine different populations (locations) using the stairway plot method (*Liu and Fu, 2015*). We chose to include in stdpopsim the **demographic model** inferred from the Gabon sample, which consists of a single population whose size fluctuated from below 80,000 (an ancient bottleneck roughly 10,000 generations ago) to the present-day estimate of over 4 million individuals (*Figure 2B*). To convert the timescale from generations to years, we used an **average generation time** of 1/11 years, as in *Miles, 2017*.

All of these parameters were set in the species entry in the stdpopsim catalog, accompanied by the relevant citation information, and the model underwent the standard quality-control process. The species entry may be refined in the future by adding more demographic models, updating or refining the recombination map, or updating the mutation rate estimates based on ones directly estimated for this species. Note that even if the mutation rate is updated sometime in the future, the demographic model mentioned above should still be associated with the current mutation rate ($\mu = 3.5 \times 10^{-9}$ mutations per base per generation), since this was the rate used in its inference.

### Bos taurus (cattle)

*Bos taurus* (cattle) was added to the stdpopsim catalog during the 2020 hackathon because of its agricultural importance. Agricultural species experience strong selection due to domestication and selective breeding, leading to a reduction in effective population size. These processes, as well as admixture and introgression, produce patterns of genetic variation that can be very different from typical model species (*Larson and Burger, 2013*). These processes have occurred over a relatively short period of time, since the advent of agriculture roughly 10,000 years ago, and they have intensified over the years to improve food production (*Gaut et al., 2018*; *MacLeod et al., 2013*). High-quality genome assemblies are now available for several breeds of cattle (e.g. *Rosen et al., 2020*; *Heaton et al., 2021*; *Talenti et al., 2022*) and the use of genomic data has become ubiquitous in selective breeding (*Meuwissen et al., 2001*; *MacLeod et al., 2014*; *Obšteter et al., 2021*; *Cesarani et al., 2022*). Modern cattle have extremely low and declining genetic diversity, with estimates of an effective population size of around 90 in the early 1980s (*MacLeod et al., 2013*; *VanRaden, 2020*;

*Makanjuola et al., 2020*). On the other hand, the ancestral effective population size is estimated to be roughly $N_e$=62,000 (*MacLeod et al., 2013*). This change in effective population size presents a challenge for demographic inference, selection scans, genome-wide association, and genomic prediction (*MacLeod et al., 2013*; *MacLeod et al., 2014*; *Hartfield et al., 2022*). For these reasons, it was useful to develop a detailed simulation model for cattle to be added to the stdpopsim catalog.

We used the most recent **genome assembly**, ARS-UCD1.2 (*Rosen et al., 2020*), with a constant mutation rate of $\mu = 1.2 \times 10^{-8}$ mutations per base per generation for all chromosomes (*Harland et al., 2017*), and a constant recombination rate of $r = 9.26 \times 10^{-9}$ recombinations per base per generation for all chromosomes other than the mitochondrial genome (*Ma et al., 2015*). With respect to the **effective population size**, it is clear that simulating with either the ancestral or current effective population size would not generate realistic genome structure and diversity (*MacLeod et al., 2013*; *Rosen et al., 2020*). Since stdpopsim does not allow for a missing value of $N_e$, we chose to set the species default $N_e$ to the ancestral estimate of $6.2 \times 10^4$. However, we strongly caution that simulating the cattle genome with any fixed value for $N_e$ will generate unrealistic patterns of genetic variation, and recommend using a reasonably detailed demographic model. Note that the default $N_e$ is only used in the simulation if a demographic model is not specified. To this end, we implemented the **demographic model** of the Holstein breed, which was inferred by *MacLeod et al., 2013* from runs of homozygosity in the whole-genome sequence of two iconic bulls. This demographic model specifies changes in the ancestral effective population size from $N_e$=62,000 at around 33,000 generations ago to $N_e$=90 in the 1980s in a series of 13 instantaneous population size changes (taken from Supplementary Table S1 in *MacLeod et al., 2013*). To convert the timescale from generations to years, we used an **average generation time** of 5 years (*MacLeod et al., 2013*). Note that this demographic model does not capture the intense selective breeding since the 1980s that has even further reduced the effective population size of cattle (*MacLeod et al., 2013*; *VanRaden, 2020*; *Makanjuola et al., 2020*). These effects can be modeled with downstream breeding simulations (e.g. *Gaynor et al., 2021*).

When setting up the parameters of the demographic model, we noticed that the inference by *MacLeod et al., 2013* assumed a genome-wide fixed recombination rate of $r = 10^{-8}$ recombinations per base per generation, and a fixed mutation rate of $\mu = 9.4 \times 10^{-9}$ mutations per base per generation (considering also sequence errors). The more recently updated mutation rate assumed in the species model ($1.2 \times 10^{-8}$ mutations per base per generation, from *Harland et al., 2017*) is thus 28% higher than the rate used for inference. As a result, if genomes were simulated under this demographic model with the species' default mutation rate they would have considerably higher sequence diversity than actually observed in real genomic data. To address this, we specified a mutation rate of $\mu = 9.4 \times 10^{-9}$ in the demographic model, which then overrides the species' mutation rate when this demographic model is applied in simulation. The issue of fitting the rates used in simulation with those assumed during inference was discussed during the independent review of this demographic model, and it raised an important question about recombination rates. Since *MacLeod et al., 2013* use runs of homozygosity to infer the demographic model, their results depend on the assumed recombination rate. The recombination rate assumed in inference ($r = 10^{-8}$ recombinations per base per generation) is 8% higher than the one used in the species model ($r = 9.26 \times 10^{-9}$). In its current version, stdpopsim does not allow the specification of a separate recombination rate for each demographic model, so we had no simple way to adjust for this. Future versions of stdpopsim will enable such flexibility. Thus, we note that genomes simulated under this demographic model as currently implemented in stdpopsim might have slightly higher linkage disequilibrium than observed in real cattle genomes. However, we anticipate that this would affect patterns less than selection due to domestication and selective breeding, which are not yet modeled at all in stdpopsim simulations.

## Conclusion

As our ability to sequence genomes continues to advance, the need for population genomic simulations of new model and non-model organisms is becoming acute. So, too, is the concomitant need for an expandable framework for implementing such simulations and guidance for how to do so. Generating realistic whole-genome simulations presents significant challenges both in coding and in choosing parameter values on which to base the simulation. With stdpopsim, we provide a resource that is uniquely poised to address these challenges as it provides easy access to state-of-the-art simulation engines and practices, and an easy procedure for including new species. Moreover, we aim

for the choices regarding the inclusion of new species to be driven by the needs of the population genomics community. In this manuscript we describe the expansion of stdpopsim in two ways: the addition of new features to the simulation framework that incorporate new evolutionary processes, such as non-crossover recombination, broadening the diversity of species that can be realistically modeled; and the considerable expansion of the catalog itself to include more species and demographic models.

We also formulated a series of guidelines for implementing population genomic simulations, based on insights from the community-driven process of expanding the stdpopsim catalog. These guidelines specify the basic requirements for generating a useful chromosome-level simulation for a given species, as well as the rationale behind these requirements. We also discuss special considerations for collecting relevant information from the literature, and what to do if some of that information is not available. Because this process is quite error-prone, we encourage wider adoption of 'code review:' researchers implementing simulations should have their parameter choices and implementation reviewed by at least one other researcher. The guidelines in this paper can be followed when implementing a simulation independently for a single study, or (as we encourage others to do) when adding code to stdpopsim, which helps to ensure its robustness and to make it available for future research. Currently, large-scale efforts such as the Earth Biogenome and its affiliated project networks are generating tens of thousands of genome assemblies. Each of these assemblies would become a candidate for inclusion into the stdpopsim catalog, although substantial changes to the structure of stdpopsim would be required to include so many distinct species. As annotations of those genome assemblies improve over time, this information, too, can easily be added to the stdpopsim catalog.

One of the important objectives of the PopSim consortium is to leverage stdpopsim as a means to promote education and the inclusion of new communities into computational biology and software development. We are keen to use outreach, such as the workshops and hackathons described here, as a way to grow the stdpopsim catalog and library while also democratizing the development of population genomic simulations in general. We predict that the increased use of chromosome-scale simulations in non-model species will lead to an improvement in inference methods, which traditionally have been quite narrowly focused on well-studied model organisms. Thus, we hope that further expansion of stdpopsim will improve the ease and reproducibility of research across a larger number of systems, while simultaneously expanding the community of software developers among population and evolutionary geneticists.

## Acknowledgements

We wish to thank the dozens of workshop attendees, and especially the two dozen or so hackathon participants, whose combined feedback motivated many of the updates made to stdpopsim in the past two years.

## Additional information

### Competing interests

Ariella L Gladstein: is an employee of Embark Veterinary, Inc. The author declares that no other competing interests exist. Jeffrey Adrion: is an employee of Ancestry DNA. The author declares that no other competing interests exist. Arjun Biddanda: is an employee of 54Gene, Inc. The author declares that no other competing interests exist. The other authors declare that no competing interests exist.

### Funding

| Funder | Grant reference number | Author |
| --- | --- | --- |
| National Science Foundation | Postdoctoral Research Fellowship 2010884 | M Elise Lauterbur |
| National Institute of General Medical Sciences | R35GM119856 | Maria Izabel A Cavassim |
| Dim One Health | RPH17094JJP | Jean Cury |

| Funder | Grant reference number | Author |
| --- | --- | --- |
| Human Frontier Science Program | RGY0075/2019 | Jean Cury |
| Brown University | Predoctoral Training Program in Biological Data Science (NIH T32 GM128596) | David Peede |
| Science for Life Laboratory | Knut and Alice Wallenberg Foundation | Per Unneberg |
| Deutsche Forschungsgemeinschaft | EXC 2064/1 - Project number 390727645 | Franz Baumdicker |
| Deutsche Forschungsgemeinschaft | EXC 2124 - Project number 390838134 | Franz Baumdicker |
| National Science Foundation | DBI-1929850 | Reed A Cartwright |
| University of Edinburgh | BBS/E/D/30002275 | Gregor Gorjanc |
| National Institute of General Medical Sciences | R01GM127348 | Ryan N Gutenkunst |
| Robertson Foundation | | Jerome Kelleher |
| National Institute of General Medical Sciences | R01HG010774 | Andrew D Kern |
| National Institute of General Medical Sciences | R35GM138286 | Daniel R Schrider |

The funders had no role in study design, data collection and interpretation, or the decision to submit the work for publication.

## Author contributions

M Elise Lauterbur, Conceptualization, Resources, Software, Investigation, Writing – original draft, Project administration, Writing – review and editing; Maria Izabel A Cavassim, Conceptualization, Resources, Software, Investigation, Writing – original draft; Ariella L Gladstein, Graham Gower, Nathaniel S Pope, Georgia Tsambos, Jeffrey Adrion, Resources, Software, Methodology; Saurabh Belsare, Arjun Biddanda, Victoria Caudill, Jean Cury, Ignacio Echevarria, Ahmed R Hasan, Xin Huang, Leonardo Nicola Martin Iasi, Ekaterina Noskova, Jana Obsteter, Vitor Antonio Correa Pavinato, Alice Pearson, David Peede, Manolo F Perez, Murillo F Rodrigues, Chris CR Smith, Jeffrey P Spence, Anastasia Teterina, Silas Tittes, Per Unneberg, Juan Manuel Vazquez, Ryan K Waples, Anthony Wilder Wohns, Yan Wong, Aaron P Ragsdale, Resources, Software; Benjamin C Haller, Gregor Gorjanc, Resources, Software, Writing – review and editing; Franz Baumdicker, Resources, Software, Methodology, Writing – review and editing; Reed A Cartwright, Resources; Ryan N Gutenkunst, Supervision, Methodology, Writing – review and editing; Jerome Kelleher, Resources, Software, Supervision, Methodology; Andrew D Kern, Software, Supervision, Methodology, Writing – review and editing; Peter L Ralph, Software, Supervision, Methodology, Project administration, Writing – review and editing; Daniel R Schrider, Methodology, Writing – review and editing; Ilan Gronau, Conceptualization, Supervision, Investigation, Methodology, Writing – original draft, Project administration, Writing – review and editing

## Author ORCIDs

M Elise Lauterbur http://orcid.org/0000-0002-7362-3618
Maria Izabel A Cavassim http://orcid.org/0000-0001-9726-1431
Ariella L Gladstein http://orcid.org/0000-0001-7735-2336
Graham Gower http://orcid.org/0000-0002-6197-3872
Georgia Tsambos http://orcid.org/0000-0001-7001-2275
Jeffrey Adrion http://orcid.org/0000-0003-1021-6000
Saurabh Belsare http://orcid.org/0000-0002-8148-1867
Arjun Biddanda http://orcid.org/0000-0003-1861-1523
Victoria Caudill http://orcid.org/0000-0002-0577-5513

Benjamin C Haller http://orcid.org/0000-0003-1874-8327
Ahmed R Hasan http://orcid.org/0000-0003-0002-8399
Ekaterina Noskova http://orcid.org/0000-0003-1168-0497
Vitor Antonio Correa Pavinato http://orcid.org/0000-0003-2483-1207
Jeffrey P Spence http://orcid.org/0000-0002-3199-1447
Silas Tittes http://orcid.org/0000-0003-4697-7434
Juan Manuel Vazquez http://orcid.org/0000-0001-8341-2390
Yan Wong http://orcid.org/0000-0002-3536-6411
Reed A Cartwright http://orcid.org/0000-0002-0837-9380
Ryan N Gutenkunst http://orcid.org/0000-0002-8659-0579
Jerome Kelleher http://orcid.org/0000-0002-7894-5253
Andrew D Kern http://orcid.org/0000-0003-4381-4680
Peter L Ralph http://orcid.org/0000-0002-9459-6866
Daniel R Schrider http://orcid.org/0000-0001-5249-4151
Ilan Gronau http://orcid.org/0000-0001-8536-4062

Reviewer #1 (Public Review): https://doi.org/10.7554/eLife.84874.3.sa1
Reviewer #2 (Public Review): https://doi.org/10.7554/eLife.84874.3.sa2
Reviewer #3 (Public Review): https://doi.org/10.7554/eLife.84874.3.sa3
Author Response: https://doi.org/10.7554/eLife.84874.3.sa4

---

# Additional files

## Supplementary files
• Transparent reporting form

## Data availability
The current manuscript is a computational study, which did not generate any new data. All source code files for stdpopsim and the species catalog are available at: https://github.com/popsim-consortium/stdpopsim, (copy archived at *PopSim Consortium, 2023*). The documentation page for the stdpopsim catalog is available at: https://popsim-consortium.github.io/stdpopsim-docs/stable/catalog.html. Figure 1 - Source code Files 1-2 and Figure 2 - Source Code File 1 contain the code and numerical data used to generate the figures.

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
