## [Editor Report · eLife assessment]

This **important** paper reports recent improvements and extensions to stdpopsim, a community-driven resource that is built on top of powerful software for performing simulations of population genomic data and provides a catalog of species with curated genomic parameters and demographic models. In addition to describing the new features and species in stdpopsim, the authors provide a set of practical guidelines for implementing realistic simulations. Overall, this **convincing** manuscript serves as an excellent overview of the utility, challenges, common pitfalls, and best practices of population genomic simulations. It will be of broad interest to population, evolutionary, and ecological geneticists studying humans, model organisms, or non-model organisms.

---

## [Referee Report · Reviewer #1 (Public Review)]

stdpopsim is an existing, community-driven resource to support population genetics simulations across multiple species. This paper describes improvements and extensions to this resource and discusses various considerations of relevance to chromosome-scale evolutionary simulations. As such, the paper does not analyse data or present new results but rather serves as a general and useful guide for anyone interested in using the stdpopsim resource or in population genetics simulations in general.

Two new features in stdpopsim are described, which expand the types of evolutionary processes that can be simulated. First, the authors describe the addition of the ability to simulate non-crossover recombination events, i.e. gene conversion, in addition to standard crossover recombination. This will allow for simulations that come closer to the actual recombination processes occurring in many species. Second, the authors mention how genome annotations can now be incorporated into the simulations, to allow different processes to apply to different parts of the genome - however, the authors note that this addition will be further detailed in a separate, future publication. These additions to stdpopsim will certainly be useful to many users and represent a step forward in the degree of ambition for realistic population genetics simulations.

The paper also describes the expansion of the community-curated catalog of pre-defined, ready-to-use simulation set-ups for various species, from the previous 6 to 21 species (though not all new species have demographic models implemented, some have just population genetic parameters such as mutation rates and generation times). For each species, an attempt was made to implement parameters and simulations that are as realistic as possible with respect to what's known about the evolutionary history of that species, using only information that can be traced to the published literature. This process by which this was done appears quite rigorous and includes a quality-control process involving two people. Two examples are given, for Anopheles gambiae and Bos taurus. The detailed discussion of how various population genetic and demographic parameters were extracted from the literature for these two species usefully highlights the numerous non-trivial steps involved and showcases the great deal of care that underlies the stdpopsim resource.

The paper is clearly written and well-referenced, and I have no technical or conceptual concerns. The paper will be useful to anyone interested in population genetics simulations, and will hopefully serve as an inspiration for the broader effort of making simulations increasingly more realistic and flexible, while at the same time trying to make them accessible not just to a small number of experts.

---

## [Referee Report · Reviewer #2 (Public Review)]

Lauterbur et al. present a description of recent additions to the stdpopsim simulation software for generating whole-genome sequences under population genetic models, as well as detailed general guidelines and best practices for implementing realistic simulations within stdpopsim and other simulation software. Such realistic simulations are critical for understanding patterns in genetic variation expected under diverse processes for study organisms, training simulation-intensive models (e.g., machine learning and approximate Bayesian computation) to make predictions about factors shaping observed genetic variation, and for generating null distributions for testing hypotheses about evolutionary phenomena. However, realistic population genomic simulations can be challenging for those who have never implemented such models, particularly when different evolutionary parameters are taken from a variety of literature sources. Importantly, the goal of the authors is to expand the inclusivity of the field of population genomic simulation, by empowering investigators, regardless of model or non-model study system, to ultimately be able to effectively test hypotheses, make predictions, and learn about processes from simulated genomic variation. Continued expansion of the stdpopsim software is likely to have a significant impact on the evolutionary genomics community.

Strengths:

This work details an expansion from 6 to 21 species to gain a greater breadth of simulation capacity across the tree of life. Due to the nature of some of the species added, the authors implemented finite-site substitution models allowing for more than two allelic states at loci, permitting proper simulations of organisms with fast mutation rates, small genomes, or large effect sizes. Moreover, related to some of the newly added species, the authors incorporated a mechanism for simulating non-crossover recombination, such as gene conversion and horizontal gene transfer between individuals. The authors also added the ability to annotate and model coding genomic regions.

In addition to these added software features, the authors detail guidelines and best practices for implementing realistic population genetic simulations at the genome-scale, including encouraging and discussing the importance of code review, as well as highlighting the sufficient parameters for simulation: chromosome level assembly, mean mutation rate, mean recombination rate or recombination map if available, effective size or more realistic demographic model if available, and mean generation time. Much of these best practices are commonly followed by population genetic modelers, but new researchers in the field seeking to simulate data under population genetic models may be unfamiliar with these practices, making their clear enumeration (as done in this work) highly valuable for a broad audience. Moreover, the mechanisms for dealing with issues of missing parameters discussed in this work are particularly useful, as more often than not, estimates of certain model parameters may not be readily available from the literature for a given study system.

Weaknesses:

An important update to the stdpopsim software is the capacity for researchers to annotate coding regions of the genome, permitting distributions of fitness effects and linked selection to be modeled. However, though this novel feature expands the breadth of processes that can be evaluated as well as is applicable to all species within the stdpopsim framework, the authors do not provide significant detail regarding this feature, stating that they will provide more details about it in a forthcoming publication. Compared to this feature, the additions of extra species, finite-site substitution models, and non-crossover recombination are more specialized updates to the software.

---

## [Referee Report · Reviewer #3 (Public Review)]

Lauterbur et al. present an expansion of the whole-genome evolution simulation software "stdpopsim", which includes new features of the simulator itself, and 15 new species in their catalog of demographic models and genetic parameters (which previously had 6 species). The list of new species includes mostly animals (12), but also one species of plant, one of algae, and one of bacteria. While only five of the new animal species (and none of the other organisms) have a demographic model described in the catalog, those species showcase a variety of demographic models (e.g. extreme inbreeding of cattle). The authors describe in detail how to go about gathering genetic and demographic parameters from the literature, which is helpful for others aiming to add new species and demographic models to the stdpopsim catalog. This part of the paper is the most widely relevant not only for stdpopsim users but for any researcher performing population genomics simulations. This work is a concrete contribution towards increasing the number of users of population genomic simulations and improving reproducibility in research that uses this type of simulations.

---

## [Author Response]

The following is the authors' response to the original reviews.

We are very glad that the editor and reviewers found our paper of broad interest to the community of population, evolutionary, and ecological genetics. We thank them for their positive feedback and insightful comments and suggestions. We have revised our manuscript to address some of the issues raised by the review. The main change we made was providing a detailed discussion of limitations of simulated genomes, focusing on considerations one needs to make when selecting a demographic model. This can be found in a new section “Limitations of simulated genomes” (pages 9-10). We made a few additional adjustments in other parts of the text based on the reviewers’ suggestions. They are all listed in the detailed point-by-point response to reviewers comments and questions below.

Editor:It was noted that demographic models (or genomic parameters) that are inferred based on certain aspects of the genomic data (eg., site frequency spectrum, haplotype structure) may not recapitulate other aspects of the data. In other words, any inferred demographic models are expected to reliably reproduce only some aspects of the genetic variation data but not necessarily all. It would be helpful to emphasize this limitation in the manuscript and to include a table summarizing the types of variation that the demographic models for the catalogued species were based on.

This is a very important point, which we addressed in the revision by adding a section entitled “Limitations of simulated genomes”. This section discusses the considerations that one should make when selecting an inferred demographic model to implement in simulation. This includes the samples used in analysis, the method used for inference, as well as various filters. In this section we also point to the documentation page of the stdpopsim catalog, which provides information about each demographic model that can help users decide whether it is appropriate for their needs. We decided not to summarize this information in a succinct table in the manuscript because it is not straightforward to summarize the strengths and potential limitations of each model in a table. Instead, we will expand the summary provided for each demographic model in the documentation page to provide additional information. See response to the second reviewer’s comment on this topic for more details.

2. It will make stdpopsim more user-friendly to include an automated module that can visualize a demographic model given the corresponding parameters (or simulation scripts).

As mentioned in the response to the first reviewer’s comment on this subject, the documentation
page of the stdpopsim catalog provides a brief summary for each demographic model, including a graphical representation. See response below for more details.

**Reviewer #1:**
In the introduction, the authors cite numerous efforts to generate high-quality reference genomes. That's not an issue in itself, but leading with this might send the message to some readers that it is these reference genome efforts that are driving the need for population genomics analysis and simulation tools, which is not really the case - why not instead give some citation attention to actual population genomics projects aiming to address the types of evolutionary questions this paper is concerned with? The reference genome citations would fit better in the section dealing with reference genomes, where they already appear.

Indeed, the desire to answer complex evolutionary questions is the main motivation for sequencing these genomes and also for generating realistic genome simulations. The reason we chose to lead with the genome-sequencing efforts is that high quality genome data is an important prerequisite for obtaining parameters for chromosome-scale simulations. So, with that perspective, these efforts which we cite are the driving force behind expansion of stdpopsim in the near future. Thus, we decided to leave these citations in the introduction. To balance things out, we now start the introduction with a statement about board questions in population genetics. Moreover, after we list the genome sequencing efforts, we added a list of specific types of questions that can be addressed by these newly emerging genomes, with relevant citations. The beginning of the introduction now reads:

“Population genetics allows us to answer questions across scales from deep evolutionary time to ongoing ecological dynamics, and dramatic reductions in sequencing costs enable the generation of unprecedented amounts of genomic data that can be used to address these questions (Ellegren, 2014). Ongoing efforts to systematically sequence life on Earth by initiatives such as the Earth Biogenome (Lewin et al., 2022) and its affiliated project networks, such as Vertebrate Genomes (Rhie et al., 2021), 10,000 Plants (Cheng et al., 2018) and others (Darwin Tree of Life Project Consortium, 2022), are providing the backbone for enormous increases in the amount of population-level genomic data available for model and non-model species. These data are being used, among other things, in inference of population history and demographic parameters (Beichman et al., 2018), studying adaptive introgression (Gower et al., 2021), distinguishing adaptation from drift (e.g. Hsieh et al., 2021), and understanding the implications of deleterious variation in populations of conservation concern (e.g. Robinson et al., 2023).”

Something that would be useful for the stdpopsim resource in general, though not necessarily something for the paper, would be some kind of more human-friendly representation of the demographic models implemented in the curated library. Perhaps I'm not looking in the right place, but as far as I can tell, if I want to study the curated demographic models, I need to go into the Python scripts on the stdpopsim GitHub page (e.g. https://github.com/popsim-consortium/stdpopsim/tree/main/stdpopsim/catalog/BosTau). Here the various parameters and demographic events are hard-coded into the scripts. To understand the model being implemented, one thus needs to go dig into these scripts - something which is not necessarily very accessible to all researchers. Visual representations, such as the one for Anopheles gambiae in Fig 2. in the paper, are more widely accessible. I wonder if such figures could be produced for all the curated models and included in the GitHub folders alongside the scripts, perhaps aided by an existing model visualization software such as POPdemog. Again, I would not suggest that this is necessary for the paper, but if practically feasible I think it would be a useful addition to the resource in the longer term.

This is a very good point. The stdpopsim catalog actually has a documentation page that provides a brief summary for each demographic model, including a graphical representation. This graphical representation is generated using demesdraw applied to the demographic model object implemented in the code. Thus, potential users do not have to dig through the Python code to figure out the details of the demographic model. We used a similar approach to generate the image of the demographic history of *A. gambiae* for Fig. 2 of the paper. The documentation page is an important part of the stdpopsim catalog, and we now added a link to it in section “Data availability”, and we mention it in key places in the manuscript, such as the caption of Fig 2.

**Reviewer #2:**
An important update to the stdpopsim software is the capacity for researchers to annotate coding regions of the genome, permitting distributions of fitness effects and linked selection to be modeled. However, though this novel feature expands the breadth of processes that can be evaluated as well as is applicable to all species within the stdpopsim framework, the authors do not provide significant detail regarding this feature, stating that they will provide more details about it in a forthcoming publication. Compared to this feature, the additions of extra species, finite-site substitution models, and non-crossover recombination are more specialized updates to the software.It would be helpful to provide additional information regarding the coding annotation (and associated distribution of fitness effects and linked selection) that is implemented in the current version of stdpopsim, but will be detailed in a forthcoming paper. This is not to take away from the forthcoming paper, but I believe this is the most important update to the software, and the current manuscript only brushes over it.

We agree that implementation of selection in simulations is a significant addition to stdpopsim. However, our intention in this manuscript is to focus on the separate effort we made in the last two years to expand the utility of stdpopsim to a more diverse set of species. We think the manuscript stands firmly even without discussing in detail the new features that allow modeling selection. The main reason we briefly mention these features in sections “Additions to stdpopsim” and “Basic setup for chromosome-level simulations” is because the released version of stdpopsim contains implemented DFEs for a few species, and we did not want to completely ignore this. We thus added a brief comment at the end of the “Basic setup” section (page 8) mentioning the three model species for which the stdpopsim catalog currently has annotations and implemented DFE models. We think that a more detailed description of how these features and how they should be used is best left to the manuscript that the PopSim community is currently writing (preprint expected later this year).

When it comes to simulating realistic genomic data, the authors clearly lay out that parameters obtained from the literature must be compatible, such as the same recombination and mutation rates used to infer a demographic history should also be used within stdpopsim if employing that demographic history for simulation. This is a highly important point, which is often overlooked. However, it is also important that readers understand that depending on the method used to estimate the demographic history, different demographic models within stdpopsim may not reproduce certain patterns of genetic variation well. The authors do touch on this a bit, providing the example that a constant size demographic history will be unable to capture variation expected from recent size changes (e.g., excess of low-frequency alleles). However, depending on the data used to estimate a demographic history, certain types of variation may be unreliably modeled (Biechman et al. 2017; G3, 7:3605-3620). For example, if a site frequency spectrum method was used to estimate a demographic history, then the simulations under this model from y stdpopsim may not recapitulate the haplotype structure well in the observed species. Similarly, if a method such as PSMC applied to a single diploid genome was used to estimate a demographic history, then the simulations under this model from stdpopsim may not recapitulate the site frequency spectrum well in the observed species. Though the authors indicate that citations are given to each demographic model and model parameter for each species, this may not be sufficient for a novice researcher in this field to understand what forms of genomic variation the models may be capable of reliably producing. A potential worry is that the inclusion of a species within stdpopsim may serve as an endorsement to users regarding the available simulation models (though I understand this is not the case by the authors), and it would be helpful if users and readers were guided on the type of variation the models should be able to reliably reproduce for each species and demographic history available for each species. It would be helpful to include a table with types of observed variation that the current set of 21 species (and associated demographic histories) are likely and unlikely to recapitulate well.

This is a very important point, which we now address in the section “Limitations of simulated genomes”, which we added to the manuscript. In this section, we expand on this topic and discuss various things that will affect the way simulated genomes reflect true sequence variation. This includes the choice of demographic inference method, but also the analyzed samples, and various filters. The main message of this section is that one should consider various things when deciding to implement a demographic model in simulation (or selecting a model among those implemented in stdpopsim). We also cite studies (including Beichman, et al. 2017), which compared different approaches to demography inference. However, we note that the conclusions of these comparisons are not as straightforward as the reviewer suggests. In particular, methods that make use of the site frequency spectrum (such as dadi) should be able to capture some aspects of haplotype structure, because this information is encoded in the demographic history. Furthermore, a demographic history inferred from a single genome (e.g., using PSMC) should do a reasonable job approximating some aspects of the site frequency spectrum. In other words, the aspects of genetic variation not modeled well by a given demographic inference method are not always predicted in a straightforward way. This is why we avoid summarizing this information in a table in the manuscript. The 2nd paragraph of the “Limitations of simulated genomes” section addresses some of these subtle considerations. In particular, we suggest that considering a demographic model for simulation requires some familiarity with the inference method and the way it was applied to data. Regarding the demographic models currently implemented in stdpopsim, we provide some information about each model in the documentation page of the catalog. When selecting a demographic model from the catalog, users should make use of this documentation to guide their decision. This is mentioned in the 3rd paragraph of the “Limitations of simulated genomes” section. Following-up on this issue, we intend to review the documentation and make sure it provides sufficient information for each demographic model. See this GitHub issue.

**Reviewer #3:**
- p5, 2nd paragraph: I think many Biologists, myself included, will think of horizontal gene transfer mostly as plasmids being transferred among bacteria and adding extra genetic material, not as homologous bacterial recombination. This made me confused about modelling horizontal gene transfer in the same way as gene conversion. It may be helpful for some readers if you specify that you are modelling this particular type of horizontal gene transfer. Some explanation along the lines of what is in Cury et al (2022) would be enough.

This is a good point. We modified the text in that sentence in the 2nd paragraph on page 5 to clarify that we are modeling non-crossover homologous recombination, and not incorporation of exogenous DNA (e.g., via plasmid transfer). The relevant part of the text now says:

“In bacteria and archaea, genetic material can be exchanged through horizontal gene transfer, which can add new genetic material (e.g., via the transfer of plasmids) or replace homologous sequences through homologous recombination (Thomas and Nielsen, 2005; Didelot and Maiden, 2010; Gophna and Altman-Price, 2022). However, the initial version of stdpopsim used crossover recombination to stand in for these processes. Although we cannot currently simulate varying gene content (as would be required to simulate the addition of new genetic material by horizontal gene transfer), the msprime and SLiM simulation engines now allow gene conversion, which has the same effect as non-crossover homologous recombination.

Following (Cury et al., 2022), we use this to include non-crossover homologous recombination in bacterial and archaeal species.”

- p5, 3rd paragraph: When you say gene conversion is turned off by default, you could refer to table 1 and briefly mention the consequence of ignoring gene conversion.

We agree that it is important to note that avoiding to model gene conversion may lead to faulty lengths of shared haplotypes across individuals. This is implied by the statement we make in the beginning of the 3rd paragraph on page 5, where we lay out the motivation for modeling gene conversion in simulation. Following the reviewer’s suggestion, we now added a statement about this in the end of that paragraph:

“Note that ignoring gene conversion may result in a slightly skewed distribution of shared haplotypes between individuals (see Table 1)”

- p7, item 1 and p9, 1st paragraph: I am not sure what you mean by genetic map here, can you define this term? I am not sure if it is synonymous with gene annotations, a recombination map, or something else. The linkage map doesn't seem to make sense to me here.

The term ‘genetic map’ referred to the recombination map whenever it was used in the manuscript. To avoid any confusion, we now removed all mentions of ‘genetic map’, and use ‘recombination map’ instead. The recombination map is relevant in item 1 of page 7 because in species with poor assemblies you will not be able to reliably estimate recombination maps, making chromosome-scale simulations less effective. In the 1st paragraph of page 9, we discuss the issue of lifting over coordinates from one assembly to another, and if you have a recombination map estimated in one assembly, you might need to lift it over to another assembly to apply it in your simulation.

- Table 1, last row, middle column: when you say "simulated population", I think it is a bit ambiguous. You mean "the true population that we are trying to simulate", but could be read as "the population data that was generated by simulation". I would delete the word simulated here.

What we mean here is that the selected effective population size should reflect the observed genetic diversity in real genomic data. We realize that the previous wording was confusing, and changed this to the following:

“Set the effective population size (Ne) to a value that reflects the observed genetic diversity”

- Figure 2, and other places when you refer to mutation and recombination rate (eg p11, last paragraph), can you include the units (e.g. per base pair, per generation)?

Throughout the manuscript, rates are always specified per base per generation. In Figure 2, this is specified in the caption (3rd line). We added units in other places in section “Examples of added species” on pages 12-13, where they were indeed missing.

- p11, "default effective population size": can you use a more descriptive word instead of the default? Maybe the historical average? Also, what is this value used for in the simulations when there is a demographic model specified (as in the case of Anopheles)?

We think that “default effective population size” is the most appropriate term to use here, since we are referring to the parameter in the species model in stdpopsim. It is correct that the value of this parameter should reflect the historical average size in some sense, but it is really unclear what this should be in the case of a species like *Bos taurus*, which experienced a very dramatic bottleneck in the recent past. We address this subtle, yet important, issue in the sentence preceding this one. If a demographic model is specified in simulation, it overrides the default effective population size, and its value is ignored (which is why we refer to it as ‘default’). We added a short sentence clarifying this in the 2nd paragraph of the “*Bos Taurus*” section (now page 12).

“Note that the default Ne is only used in simulation if a demographic model is not specified.”

- p8, when you say "Such simulations are useful for a number of purposes, but they cannot be used to model the influence of natural selection on patterns of genetic variation.": You may want to bring up the discussion that many of these neutral parameters taken from the literature could have been estimated assuming genome-wide neutrality, and thus ignoring the effect of background selection. Therefore the parameter values might reflect some effect of background selection that was unaccounted for during their estimation.

This is an important subtle point, which we now address in the section “Limitations of simulated genomes”, which we added to the revised manuscript. In that section, we discuss various limitations of simulations, focusing on inferred demographic models. We address the potential influence of the segments selected for analysis toward the end of 2nd paragraph in that section (page 9):

“... all methods assume that the input sequences are neutrally evolving. This implies that technical choices, such as the specific genomic segments analyzed and various filters, may also influence the inferred model and its ability to model observed genetic variation.”

Interestingly, background selection in itself typically does not have a strong effect on the inferred model. This is something that is examined in the forthcoming publication that presents simulations with natural selection in stdpopsim.

- Why are some concepts written in bold (eg effective population size, demographic model)? Were you planning to make a vocabulary box? I think this is a good idea given that you are aiming for a public that can include people who are not very familiar with some population genetics concepts.

In the “Examples of added species” section, we use boldface fonts to highlight the model parameters that were determined for each species. We added a statement clarifying this in the beginning of this section (page 11), and made sure that all the relevant parameters were consistently highlighted throughout this section. In other sections, we use boldface fonts only for titles. A few cases that did not conform to this rule were removed in the current version. We did not intend on adding a vocabulary box, but considered this when revising the manuscript, due to the reviewer’s suggestion. However, we found it difficult to converge on a small (yet comprehensive) set of terms with accurate and succinct definitions. We think that the important terms are adequately defined within the text of the manuscript, providing sufficient information also for readers who are not expert population geneticists.

- p4, 2nd paragraph: Are these automated scripts that are used to compare models publicly available? If you are suggesting that people use this approach generally when coming up with a simulation model (p8, penultimate paragraph), it would be helpful to have access to these automated scripts.

The scripts are part of the public stdpopsim repository on GitHub, and may be used by anyone. Some components of these scripts are more easy to apply in general, such as comparing a demographic model with one implemented separately by the reviewer. This step, for example, is achieved by application of the Demography.is_equivalent method in msprime. Other parts of the comparison depend on the specific structure of python objects used by stdpopsim, so they are not likely to be useful when implementing simulations outside the framework of stdpopsim.

- p9, 1st paragraph, and p.12 2nd paragraph: instead of adjusting the mutation rate to fit the demographic model (and using an old estimate of the mutation rate), would it be ok to adjust the demographic model to fit the new mutation rate? E.g. with a new mutation rate that is the double of a previous estimate, would it be ok to just divide Ne by 2 such that Ne*mu is constant (in a constant population size model)? I imagine this could get complicated with population size changes.

In principle, this could be done if you were simulating neutrally evolving sequences (without modeling natural selection). Since the coalescence is scale-free, then you can scale down all population sizes and divergence times by a multiplicative factor, and scale up migration rates and the mutation rate by the same factor, and you get the exact same distribution over the output sequences. However, making sure you get the scaling right is tricky and is quite error-prone. Especially considering the fact that you have to do this every time the mutation rate of a species is updated. Moreover, once you start modeling natural selection, this scale-free property no longer holds. Thus, the simple solution we came up with in stdpopsim is to attach to each demographic model the mutation rate used in its inference.